# Pre- and postmortem tyrannosaurid bite marks on the remains of *Daspletosaurus* (Tyrannosaurinae: Theropoda) from Dinosaur Provincial Park, Alberta, Canada

DWE Hone[1] and DH Tanke[2]

[1] School of Biological and Chemical Sciences, Queen Mary, University of London, London, UK
[2] Royal Tyrrell Museum, Drumheller, Alberta, Canada

## ABSTRACT

Trace marks on the bones of non-avian dinosaurs may relate to feeding by large carnivores or as a result of combat. Here the cranium and mandible of a specimen of *Daspletosaurus* are described that show numerous premortem injuries with evidence of healing and these are inferred to relate primarily to intraspecific combat. In addition, postmortem damage to the mandible is indicative of late stage carcass consumption and the taphonomic context suggests that this was scavenging. These postmortem bites were delivered by a large bodied tyrannosaurid theropod and may have been a second *Daspletosaurus*, and thus this would be an additional record of tyrannosaurid cannibalism.

## INTRODUCTION

Evidence of carnivore-consumed interactions in the fossil record are important for reconstructing the ecology and determining the trophic interactions of extinct taxa. Much research has investigated the diet and putative behaviours of the carnivorous members of the theropod clade of non-avian dinosaurs (hereafter simply 'dinosaurs'). These have examined how prey may have been acquired (*Rayfield, 2005*), handled (*Fowler et al., 2011a*) and consumed (*Erickson & Olson, 1996*), and what prey was targeted (*Hone & Rauhut, 2010*). Theropods had a diverse diet including fish (*Charig & Milner, 1997*), mammals (*Larsson et al., 2010*), lizards (*Ostrom, 1978*), pterosaurs (*Hone et al., 2012*), other non-avian dinosaurs (*Varricchio, 2001*) and birds (*O'Connor, Zhou & Xu, 2011*). Evidence suggests at least some theropods were both predators and scavengers, and in particular, large tyrannosaurs were both predators and scavengers (*Holtz, 2008*; *Hone & Watabe, 2010*; *DePalma et al., 2013*).

Three main lines of evidence are typically used to show trophic interactions between carnivorous theropods and consumed taxa: gut contents of ingested bones or other elements, bite marks on material that was not ingested, and coprolites of formerly consumed material. Both stomach contents and coprolites for theropod dinosaurs are

Corresponding author
DWE Hone, d.hone@qmul.ac.uk

known but extremely rare (*Chin, 1997*), and even bite marks are uncommon (though in part likely to be under recorded).

Note that here we use the term "carnivore-consumed" as opposed to the more common "predator–prey" as the latter makes an implicit statement that the consumed animal was actively hunted and killed, and thus is inappropriate for referring to scavenged food items, or where the status is not known. Therefore, "predator–prey" should be restricted in use to instances where it can be determined that prey was actively killed by the consuming carnivore in question or this was a likely intended outcome of the interaction (*DePalma et al., 2013*).

In general, theropods may have fed carefully and avoided tooth-on-bone contact (*Hone & Rauhut, 2010*), but tyrannosaurs with their powerful bites (*Rayfield, 2004*) may have been exceptions (*Hone & Rauhut, 2010*). Certainly, tyrannosaurs were capable of both huge and powerful bites into bone (*Erickson & Olson, 1996*) but also were selective feeders, adjusting biting style to the material at hand and the intended results (*Hone & Watabe, 2010*). Even so, overall derived tyrannosaurs seem to have produced more bite marks than other theropods (*Jacobsen, 1998*) suggesting that their feeding strategy involved greater tolerances of tooth-bone contact and/or actual deliberate biting of, and potentially regular consumption of, bones.

Increasing numbers of theropod-theropod carnivorous interactions are known (e.g., *Sinocalliopteryx*—*Xing et al., 2012*; *Tyrannosaurus*—*Longrich et al., 2010*) and while likely relatively rare (if only because carnivores are much less common in macroscopic terrestrial ecosystems than are herbivores) show that theropods did consume, and probably on occasion actively killed, other carnivores for food. However, bites or injuries to theropods may not be the result of attempted predation alone. Some large theropods engaged in cranio-facial biting (*Tanke & Currie, 1998*; *Peterson et al., 2009*; *Bell & Currie, 2010*), presumably at least in part being some form of ritualized combat linked to socio-sexual dominance contests, and combat may also have involved wounds inflicted by the feet (*Rothschild, 2013*). Healed bites on the crania of large theropods show evidence of combat with other large theropods. These are not normally matched with injuries elsewhere (which are comparatively uncommon) on the body of the animals suggesting it was not a predation attempt (cf. damage distal parts of the axial series when predation was attempted: *Carpenter, 2000*; *DePalma et al., 2013*). This adds to the complexity of correctly determining interactions between two large carnivores from potential predation/consumption of one by another.

Here we document the remains of an immature tyrannosaurine theropod, *Daspletosaurus* sp. that shows evidence of numerous healed injuries to the cranium and mandible inflicted by another large theropod, probably a tyrannosaurid. Additional bite marks inflicted on the mandible appear to be post mortem and are attributed to another tyrannosaurid, and evidence for the decay and disarticulation of the material at the time these marks were inflicted suggests this was a scavenging event.

## Locality information

The specimen was originally discovered by PJ Currie in the 1994 field season and was excavated from Quarry 215 (locality L0315), Dinosaur Provincial Park, Alberta, Canada, over several summers. The specimen is from the lower part of the Dinosaur Park Formation.

Despite disarticulation, it can be seen from the general pattern of mapped bones that the carcass arrived on site more or less intact (distal tail to tip of snout, ribs, pelvis), lying on its right side in opistotonic posture, and then decayed with the elements coming apart in place. The position of the mapped bones versus the edge of the quarry suggests that is quite possible that the major fore- and hindlimb elements were present but eroded away, though no eroded limb bone pieces were found aside from part of one femur. Erosion rates and postulated position of the limbs suggests that if this occurred it was many years before the discovery of the specimen.

## Attribution to *Daspletosaurus*

Identifying TMP 1994.143.0001 as *Daspletosaurus* is not straightforward. *Currie (2003)* described this material as a juvenile of *Daspletosaurus*, but did not give a reason for this assignment (although he noted much in common with confirmed specimens of this genus). *Holtz (2004)* noted that *Daspletosaurus* could be diagnosed by a convex tab-like process on the postorbital, an element not preserved here. Some of the diagnostic characteristics originally given by *Russell (1970)* are problematic or no longer diagnostic for the genus (e.g., "maxillary teeth large, not greatly reduced posteriorly") although others are present here (e.g., "premaxilla does not contact nasal below external naris"). The specimen appears to have 18 dentary alveoli, one higher than is normal for *Daspletosaurus*, but also higher than any other tyrannosaurid (except occasional specimens of *Albertosaurus*—*Carr & Williamson, 2000*) supporting this assignment. Furthermore, *Carr & Williamson (2004)* noted that *Daspletosaurus* is unusual for a tyrannosaurid in having the tooth denticles of the mesial carinae reach the base of the tooth, a feature seen here. *Carr, Williamson & Schwimmer (2005)* provided two unique attributions to the lacrimal that appear to be present in TMP 1994.143.0001—an apex is present on the corneal process, dorsal to the ventral ramus, and the dorsal ramus of the lacrimal is inflated. Finally, coding of the analysis by *Brusatte et al. (2010)* also suggests that the character "maxillary fenestra, location abutting the ventral margin of the antorbital fossa" (their character 18) is diagnostic for this genus compared to other tyrannosaurines and this is apparently present here although there is a slight space between the two margins. Collectively, despite the issues described above and the non-adult status of the animal, this specimen can be confidently assigned to this genus.

## Preservation of material

The specimen (TMP 1994.143.0001) is mostly very well preserved, with superb surface texture and fine details preserved. However, there is a great deal of cracking on the cortex of many elements (and especially on the skull) and some pieces show some heavy damage (e.g., the partial femur). There appears to be little distortion to the preserved bones, even

in relatively thin and fragile skull elements, though the skull piece does seem to have been sheared somewhat. The bones are a rich brown colour and lie within a grey–green coloured, fine-grained and silty matrix with small clayballs.

Some pieces of the carcass were still articulated when discovered (e.g., two cervicals, several dorsals and some dorsal ribs) indicating limited transport, and there is no evidence of fluvial abrasion to bone surfaces across the specimen as a whole. Even small and fragile elements or parts were in excellent condition and unabraded (e.g., braincase, zygopophyses, gastralia). Some broken pieces which were separated from the skull could be restored perfectly to their original positions shows a lack of abrasion or wear to the edges, and that the energy in the environment was insufficient to transport them far or abrade them. This inference of a low energy environment is further supported by the lack of wide scattering of elements, and the lack of alignment of long-axis orientations of bones

The skeleton is rather incomplete (see below); however, the evidence for recent erosion suggests that more material may have been lost prior to its discovery and excavation. There is some stratification of elements of the tyrannosaur within the quarry, with pieces lying over each other and separated by sediment, suggesting possible transport or the movement of elements by other agents (e.g., scavengers).

Elements of vertebrates that do not belong to the *Daspletosaurus* were also recovered from the quarry. These include a tibia and phalanx of small theropod, a very incomplete hadrosaur femur, ornithischian teeth, a single osteoderm and teeth from crocodilians, a piece of trionychid turtle plastron, a *Myledaphus* tooth, cf. *Champsosaurus* tooth, and finally a salamander vertebra. All are typical components of the Dinosaur Park fauna and occur regularly in the quarries (*Eberth & Currie, 2005*).

## Description

The specimen TMP 1994.143.0001 is that of an osteologically immature individual of *Daspletosaurus* sp., a large tyrannosaurine theropod. The animal is represented by a mostly complete, but disarticulated, skull and partial postcranium. Most of the skull material has previously been illustrated (*Currie, 2003*), and much work has been done on tyrannosaurine cranial morphology, ontogeny and taxonomic implications (e.g., *Brochu, 2003*; *Hurum & Sabath, 2003*; *Carr & Williamson, 2004*; *Brusatte, Carr & Norell, 2012*; *Hone et al., 2011*). Therefore, the material will not be redescribed here, and instead we limit ourselves to comments on the elements of interest and damage or marks to them.

A large piece of the skull consisting of the front and right part of the cranium is represented by the premaxillae, maxillae, nasals and the right jugal and lacrimal. On the right side, the cranium is intact as far as the orbit (Fig. 1), and the left side is complete as far as the anterior part of the antorbital fenestra (Fig. 2). This part of the cranium as preserved totals 550 mm in length. Numerous other cranial and mandibular elements are also preserved including fragile and/or rarely preserved elements such as the braincase, supradentary, ectopterygoids, surangular, and palatines. There are alveoli for four premaxillary teeth and 13 maxillary teeth. All but one of the premaxillary teeth are preserved in the right side of the skull, with just one premaxillary tooth and six maxillary

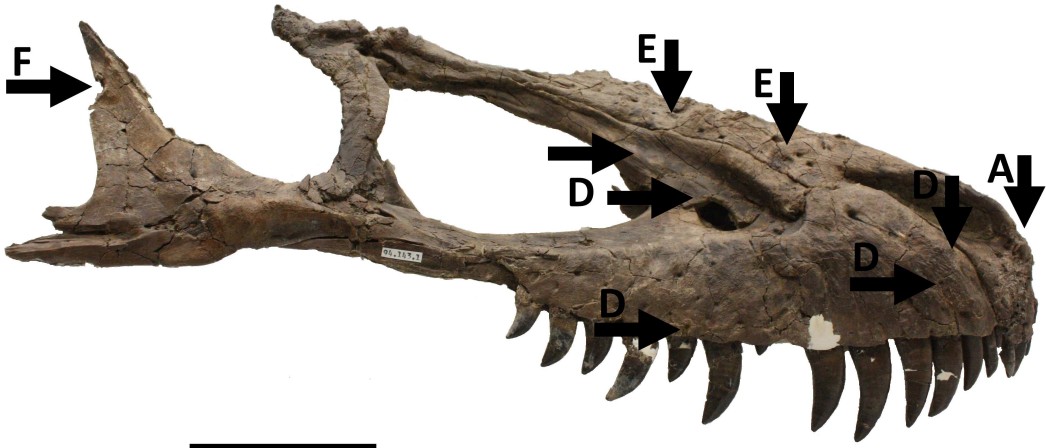

**Figure 1  Skull in right lateral view showing numerous injuries indicated with black arrows and the relevant code letter (see the text for details).** See also Figs. 4 and 5 for additional details on the right maxilla not indicated on this figure for clarity. Scale bar is 100 mm.

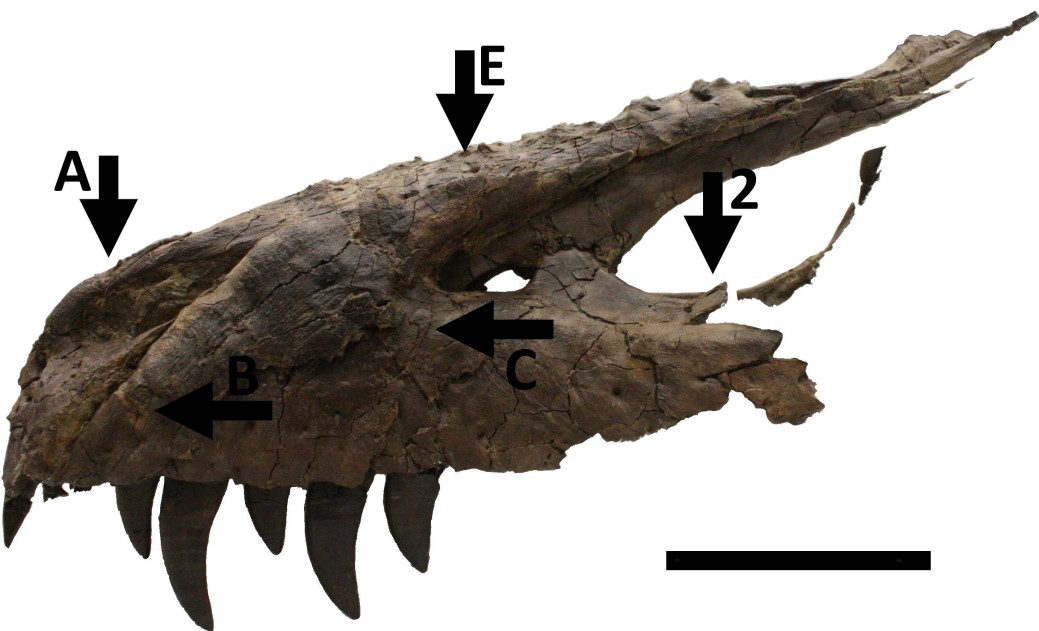

**Figure 2  Skull in left lateral view showing numerous injuries indicated with black arrows and the relevant code letter or number (see the text for details).** Note that for this view the 'floating effect' of a thin lacrimal and other posterior elements is due to the fact that these are supported by plaster which has been digitally edited out so not all of the medial side of the bone can be seen. Scale bar is 100 mm.

teeth on the left and the others being missing. There may have been more originally in the maxilla, but the posterior part on the left is broken. The largest maxillary teeth have a crown height of just over 50 mm.

The main piece of the cranium is slightly distorted, showing shearing along the midline of the skull, and elements have shifted and separated slightly along the sutures. Numerous
foramina are present on the premaxilla, along the ventral margin of the maxilla and along the nasals. The nasals are especially rugose and show peaks and pits across their dorsal surface.

The right dentary is nearly complete, with nine preserved teeth plus one emerging replacement crown and a total of 18 dental alveoli. The dentary teeth are displaced from the jaws and mostly do not lie in their respective alveoli. The total length of the mandible as preserved is 380 mm. There is no sign of the left dentary, but this must have originally been present based on the number of isolated teeth recovered (see 'Discussion' for details).

At least parts of most major areas of the postcranial skeleton are preserved, including cervical vertebrae, dorsal vertebrae, dorsal ribs and gastralia, caudal vertebrae and chevrons (the caudal vertebrae and chevrons are in especially good condition), a partial ilium and partial femur. However, most have suffered some damage and are incomplete, with severe crushing damage to some elements (e.g., the femur).

The specimen is considered to be a non-adult animal based on the incomplete fusion of a number of cranial and vertebral elements. Although apparently varied in at least some tyrannosaurs (*Fowler et al., 2011b*), fusion in archosaurs is generally considered to being in the distal part of the vertebral column and proceed anteriorly. Several articulated posterior cervicals in this specimen plainly show a suture between the neurocentral arch and the centrum, and an isolated cervical vertebra (probably the atlas or axis) shows separation between these two parts. In addition, one mid-dorsal vertebra appears to be completely fused while a number of proximal caudals show a suture line, but the degree of fusion (if any) cannot be determined. None of the well-preserved distal caudal vertebrae show any trace of a suture. The nasals are also fully fused to one another which occurs early in tyrannosaur ontogeny, suggesting the animal is not very young (*Tsuihiji et al., 2011*) and fusion in the braincase is also an indicator of maturity (*Sereno et al., 2009*) and only limited fusion of some braincase parts (see *Currie, 2003*) suggests this animal was neither very young nor osteologically mature. Collectively this pattern of fusion of various elements implies incomplete osteological maturity for the animal.

The overall intermediate size of the individual is also indicative of incomplete growth. For example, an adult specimen of *Daspletosaurus* has frontals of around 145 mm compared to just 99 mm here (data from *Currie, 2003*). Similarly, adult specimens have a dentary tooth row of 430–455 mm (*Currie, 2003*) compared to that of TMP 1994.143.0001 which is only 290 mm. *Currie (2003)* estimated the total length of TMP 1994.143.0001 at 5.8 m based on the size of the preserved parts of the skull and it was estimated at 496 kg and to be ten years of age by *Erickson et al. (2004)*.

## Bite marks and breaks

Every available element and fragment of bone (which totalled over a hundred pieces) was examined closely for bite marks or traces of damage. All damage that could be identified as being of Cretaceous in age, rather than more recent breaks or erosion, was restricted to the cranial and mandibular elements (including the teeth) with the exception of a healed break on a dorsal rib. There is also no indication of bite-mark-like damage to any other

vertebrate material in the quarry. Both damage that was premortem and postmortem could be identified, in addition to some of indeterminate origin.

### Premortem

The vast majority of the identified marks on the skull are premortem, indicating that numerous injuries or infections occurred at various times in the life of the individual. These must also have occurred prior to death at such a time as to allow for evidence for healing and repair to be visible. Premortem damage is considered as such based on evidence of healing through pathology, or presence of finely pitted reactive bone and also anomalous directions of pits combined with raised ridges (*Rothschild & Martin, 1993*; *Tanke & Rothschild, 2002*).

A. A bite on the tip of the snout with bulged and pathological area on the right ascending process of the premaxilla, and apparent fusion of the right and left premaxillary processes (Figs. 1–3). This damage is associated with a small subcircular mark approximately 13 mm in diameter, and 6 mm in depth, on the anteriolateral face of the left premaxilla. A second suboval (16 mm long by 6 mm across, and less than 2 mm deep) mark lies 6 mm posteriorly to this at the juncture of the right premaxilla and right nasal. The long axis of the oval is subparallel to the ramus of the premaxillary process.

B. A lesion which is close in form to a bite-and-drag mark (anterior to dorsally directed) on the anterolateral part of left maxilla (Fig. 2). This is 22 mm long by 8 mm (widest point) and maximum depth of the puncture is 1.5 mm. It is a distinctly different colour from the surrounding bone, being more of a burnt-orange than brown.

C. Comma-shaped damage anteroventral to the promaxillary fenestra on the right maxilla (Fig. 2). This is 22.5 mm long, 6.5 mm wide at the top and 1.5 mm wide at base. It appears to be a bone avulsion that was torn out, and is recessed in the excavation of the maxilla. There is evidence of healing on the maxilla around the edges of this damage, but not apparently on the avulsion. This is close in appearance to some tyrannosaurine bite-and-drag marks (sensu *Hone & Watabe, 2010*), and here is anteroventrally directed.

D. A series of lesions and injuries on the right maxilla (Fig. 1). Two very small lesions lie on the posteriorly directed ramus of the right maxilla, below the right nasal. The first is of mild osteomyelitis and tracks the upper margin of the right promaxillary fenestra (Fig. 4). This is a small subcircular lesion, 7 mm long and 6 mm tall and 2 mm deep. The second lies in line with the long axis of the left naris, and is a large and subcircular lesion measuring 15 mm in diameter and with a maximum depth of 2 mm. This lies alongside the edge of the maxilla, but does not extend onto the other element.

Just dorsal to the seventh maxillary alveolus is a groove that is posteroventrally directed, it is 16.5 mm long is up to 6.5 mm wide and 2 mm deep (Fig. 5). Similarly, on the ventral margin of maxilla above ninth alveolus is a dorsoventrally directed notch 11 mm tall, 4 mm in width and up to 2.75 mm deep.

A pair of conjoined subcircular depressions are apparent on the anterodorsal edge of the right maxilla, close to the suture with the right premaxilla. A major lesion lies on the anterior part of the right maxilla, represented by a moderately thickened patch (slightly

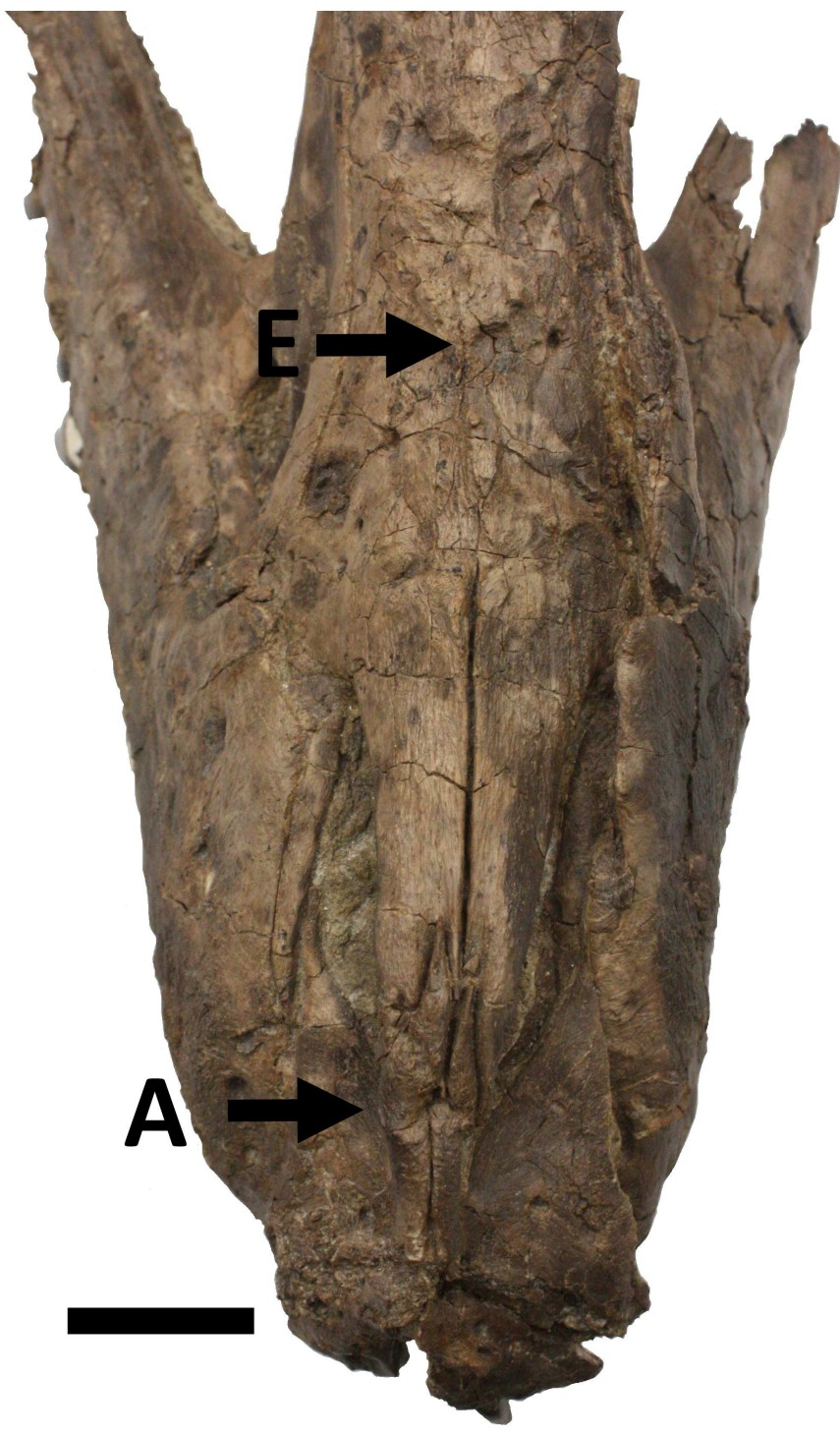

**Figure 3 Snout in dorsal view showing damage A to the ascending processes of the premaxillae and E, a large subcircular lesion in the nasals.** Scale bar is 50 mm.

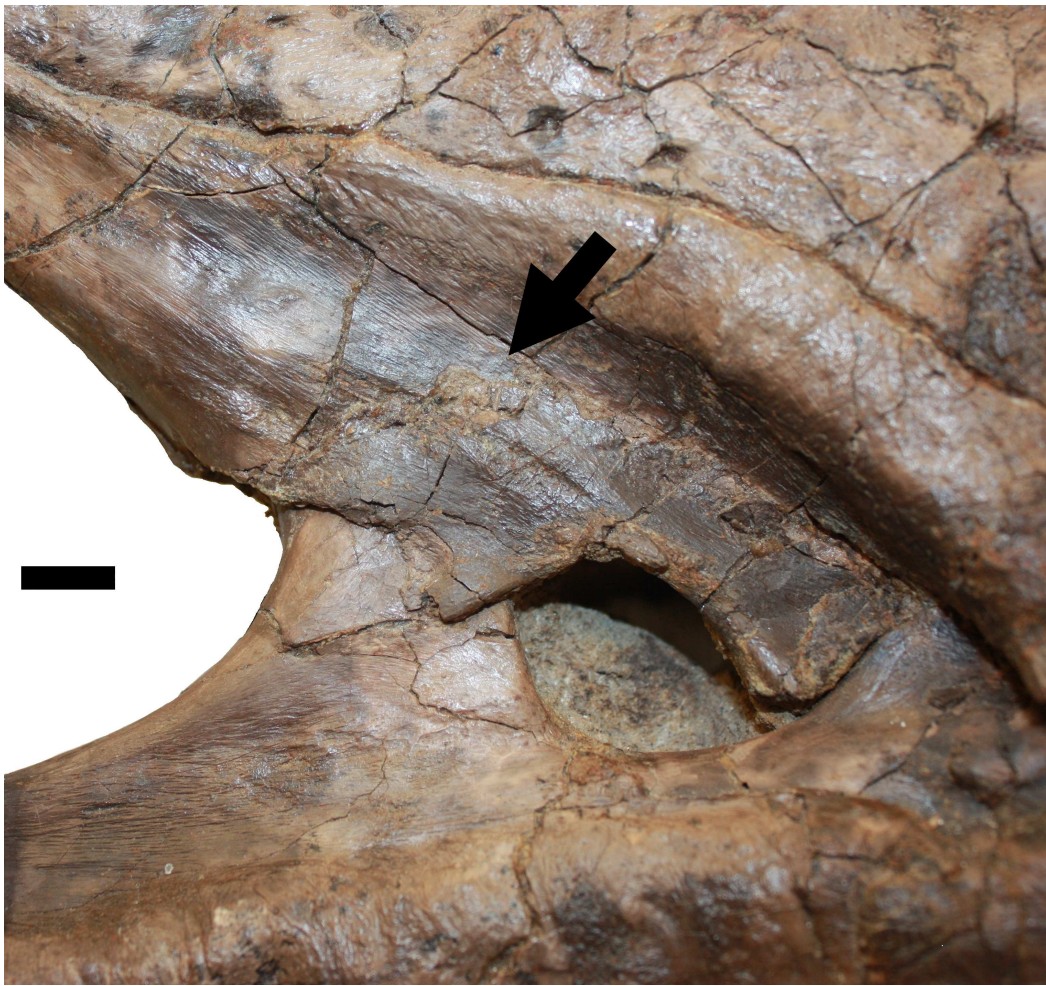

**Figure 4 Lesion associated with the proxmaxillary fenestra.** Lesion above the promaxillary fenestra described in D (black arrow). Scale bar is 10 mm.

raised by 1 or 2 mm compared to the surrounding bone) of reactive bone that is suggestive of osteomyelitis. This area measures 48.5 mm tall, and is up to 31 mm wide.

E. A large subcircular puncture (9 mm across, approximately 2 mm deep) on the dorsal part of the middle of the left nasal (Fig. 3). The right nasal also has small lesion (10 mm long by 5 mm) just dorsal to the posterodorsal edge of the narial opening. A second lesion occurs on the right side of the right nasal where it contacts the maxilla, lying 30 mm posterior to the posteriormost part of the right naris (Figs. 1 and 2). This lesion is subcircular and 14 mm in diameter, and around 1 mm in depth. It is composed of darker bone colour than the surrounding tissue and is finely pitted both inside the lesion and also around its margin. There are two slightly raised irregularly shaped 'islands' of bone that sit within the lesion, these are prominent and have a smoother texture than the other pitted bone.

F. A lesion on the right jugal that penetrates the bone fully (Fig. 1). A semicircular area which is estimated at 11 mm tall and approximately 7 mm across (based on the lack of a posterior part) is missing from the element. Mildly reactive bone is present around the

edge of this lesion, increasing the overall length of the damage to 24 mm by 14.75 mm, with an anteriorly a swollen area showing evidence of osteomyelitis. A small premortem lesion, ventral to the above described one, is also present and measures 11 mm long by 8.5 mm tall.

G. A pit on the left lacrimal is 6.5 mm across and 3 mm deep. See also below.

H. A posteriorly directed lesion, that includes a pit in the anterior part, that lies on the left posterior edge of the postorbital descending process, just above the tip of the ascending process of the left jugal (Fig. 6). The lesion is 2 mm deep, 13 mm long and 5 mm wide and shows signs of healing, with the medial face shows swelling, and is invaginated into the cavity. Medially, there is a slight overlap of the fragments that make up the lesion showing that after the break occurred and before or during healing one part slipped over the other, prior to being fused into their current positions.

I. A deep and elongate lesion with a rimmed margin on the right postorbital (Fig. 6). It lies halfway down the anterior edge of descending process. The lesion is 18.5 mm in length and 9 mm wide (including the raised edges) and with a maximum depth of 2.5 mm. The surface of the lesion shows mildly reactive bone.

J. There is a light patch of osteomyelitic bone, diagnosed by texture changes, present on the left quadratojugal, in the middle of the element.

K. Damage to both sides of the nuchal crest. When seen in posterior view, the nuchal crest is an unusual shape and apparently the two sides have both been damaged though in different ways (Fig. 7). Although partly broken, a natural, finished bone surface is present on much of the dorsal margins. The right side has a semicircular section missing, lined by normal cortical bone, and suggestive of a hole in the flange. This is similar in form to an injury seen in a specimen of *Troodon* that may have been a cyst (*Currie, 1985*) or a tooth puncture wound (*Tanke & Rothschild, 2002*). On the left side of the crest, the dorsal edge curves very rapidly in a ventral direction and, although now broken, would not have met the rising ventrolateral margin. Again however, the edges are natural, suggesting the shape is mostly genuine and not the result of breakage, though this is considered to be likely be pathological or trauma induced. There is only limited distortion elsewhere in the element, suggesting again the feature is probably genuine.

L. Damage to the lateral margins of the dentary (Fig. 8). These are present mostly on the ventral side and this element, and these consist of subcircular puncture wounds, or pairs of elongated ridges that are indicative of former scores. One point on the extreme ventral surface is barely visible in lateral or medial view, and is represented by a slight bump associated with a slight score. An additional puncture lies on the lateral side of the right dentary, ventral to the meeting point between the sixth and seventh alveoli. This mark is 4 mm tall by 5 mm long and 1 mm in depth. There is a small rectangular piece of bone that lies on the anterior edge of the mark that appears to have been lifted up (an avulsion) and settled back into place and subsequently fused back with the bone.

M. Two holes, and a large patch of ostomyelitis are on the right surangular. This ostomyelitic bone is stronger anteriorly, and then fades towards the mandibular fenestra. Additional osteomyelitic bone lies above and posterior to the fenestra and there is still more

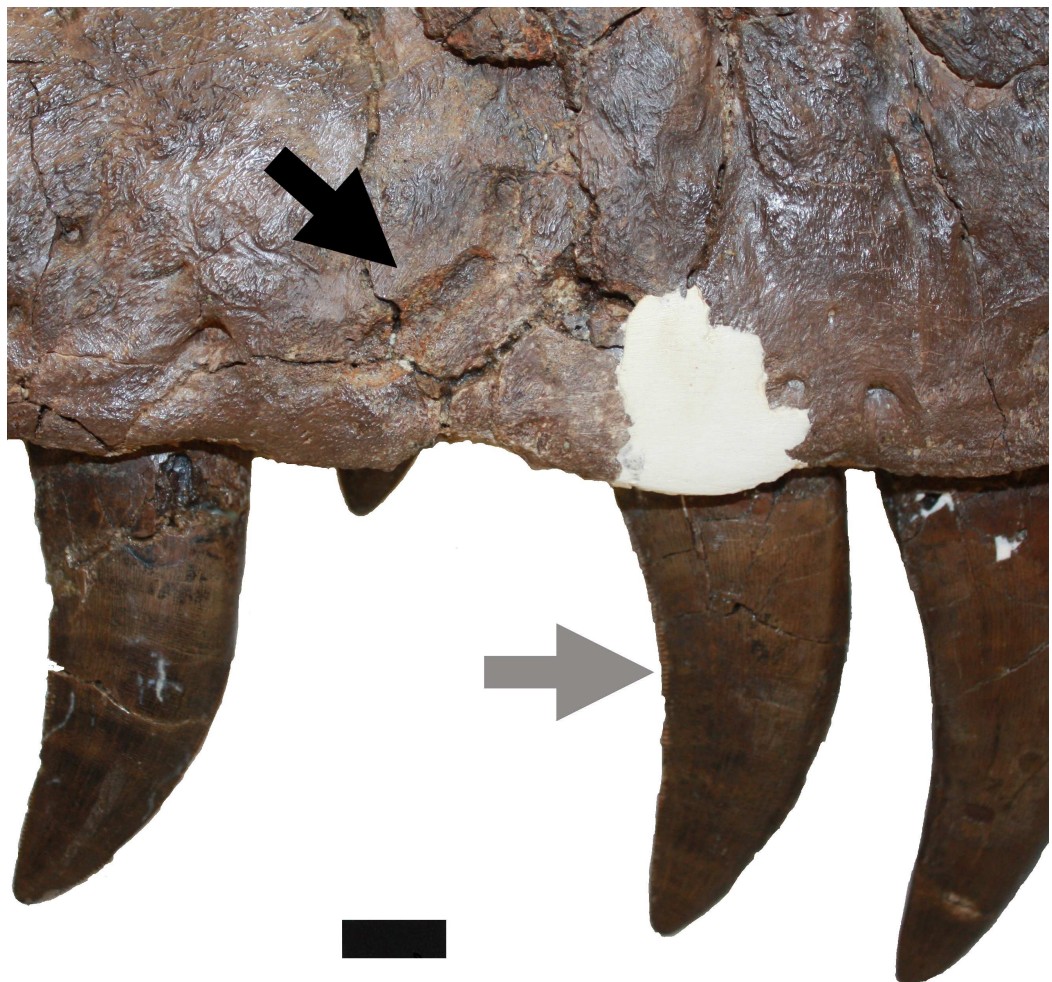

**Figure 5  Damage on the right maxilla.** Damage to the right maxilla described in D (upper black arrow). Also indicated is the damage to the posterior carina of the maxillary tooth described in N (grey arrow). Scale bar is 10 mm.

along the dorsal edge of this element, including small 'hump' on the posterior part of the surangular (Fig. 10). Two damage points from preparation and/or excavation also lie on the right surangular

N. The posterior carina of fifth right maxillary tooth has suffered serious ablation (Fig. 5). This damage is here attributed to occlusion wear with the dentary tooth row (*Schubert & Ungar, 2005*). The first and third teeth of the right maxilla both have their tips broken and with subsequent wear leading to smoothing of the affected areas. The eleventh tooth of the right dentary has also suffered damage which is interpreted here as being a result of premortem malocclusion.

   A number of other very fine marks and cracks are present in places across the cranium. These may represent possible osteopathic marks and traces, but could be taphonomic artefacts based on the partial separation of material and the dorsoventral collapse of the nasals and other parts. As they are minor it is difficult to determine their origins, and these

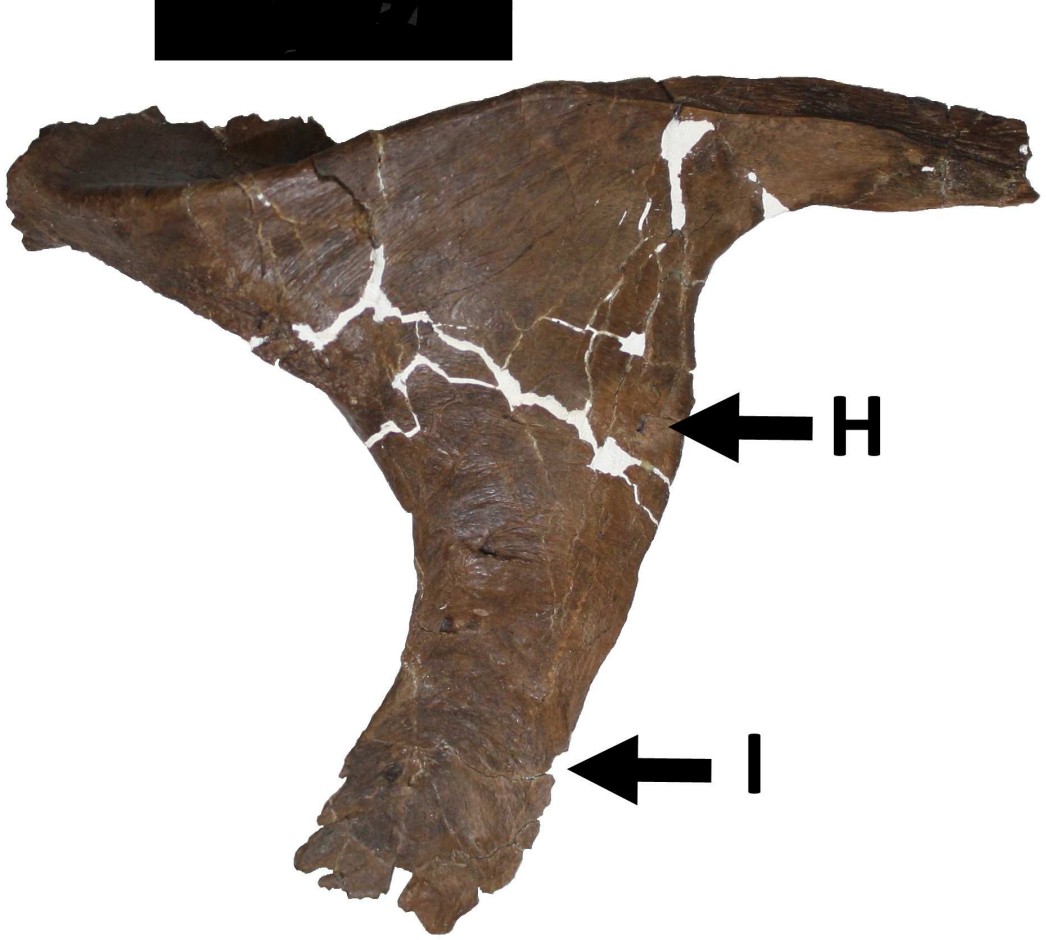

**Figure 6 Damage to the postorbital.** Damage H and I to the left postorbital. Scale bar is 50 mm.

are not discussed further. In addition to the cranial damage, one dorsal rib shows evidence of having been fractured, but this was well healed at the time of death.

### Postmortem

Postmortem damage on the material is assessed based on the lack of any indications of healing (swelling, reactive bone etc.) with tooth-marks being identified as being sub-parallel traces damaging the bone cortex. One clear set of bite-marks appears on the medial face of the posterior part of the right dentary. A number of other elements show possible evidence of biting with breaks aligning between elements or with localised impacts on bone.

i. Four tooth marks on posteromedial part of dentary are considered to be the result of biting (Figs. 11 and 12). These are well spaced (around 15–20 mm between each) and light—they barely graze the surface of the bone. There is no evidence of healing of these implying they were inflicted either postmortem, or very shortly before death. However, the position of these marks on the medial face of the extreme posterior part of the dentary suggests they were unlikely to be premortem.
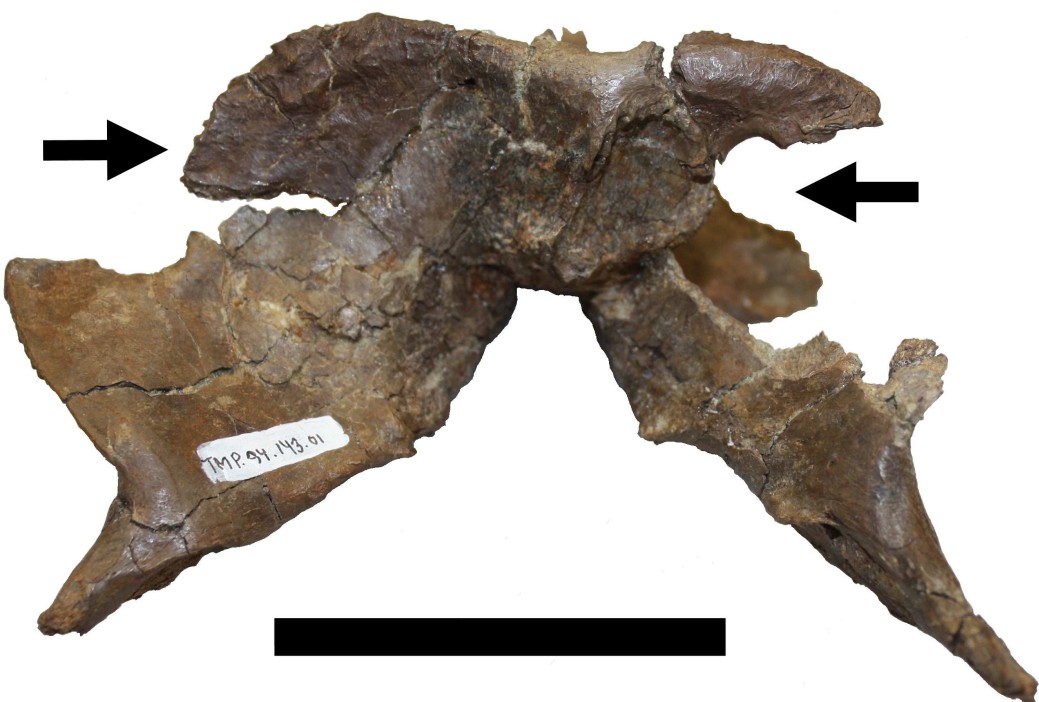

**Figure 7 Damage to the occipital region.** Damage K to the two sides of the occipital region (in posterior view). On the left side the dorsolateral part has been removed, and on the right, a large subcircular puncture has penetrated the bone. Scale bar is 50 mm.

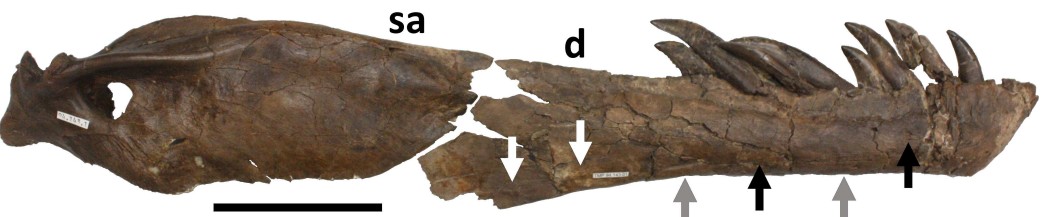

**Figure 8 Restoration of right mandible.** Approximate reconstruction of the lateral elements of the right mandible with damage L to the lateral face and to the ventral margin of the dentary indicated. Punctures are indicated with black arrows, ridges on the lateral face of the bone are in white arrows and grey arrows indicate the location of marks on the underside of the bone. Abbreviations: d, dentary; sa, surangular. Scale bar is 100 mm.

ii. A dislodged bone fragment lies between the two posteriomost alveoli preserved in the right dentary. This fragment of bone has apparently been forced down into the space between the alveoli (Figs. 11 and 12). Given that the original position of the dentary as recovered was inverted, this suggests that a strong, but localised, impact broke and drove this fragment into this position. It is therefore indicative of a bite, indeed probably the same one that delivered the sub-parallel damage described above in section (i). We consider this damage unlikely to be the result of trampling given that the element is otherwise intact, and also not likely due to fluvial action given the relative positions of the dentary and cranium.

iii. Breaks across the posterior parts of the right dentary. As preserved these are considered to be likely the result of a bite. The edges of the breaks are rough and indicative of a break of green bone, and are roughly in line with the series of four scores on the dentary described above (i). The lack of fluvial damage to the rest of the specimen suggests this is an inflicted injury. Similarly, the extreme dorsal anterior part of the surangular is damaged and may be linked to the bite at the posterior part of the dentary. Part of the dentary posterior to this point has separated from the rest of the element and is now lost; however, there are photographs of this part prior to this loss showing the original condition of the material (Fig. 12).

iv. Broken process on the right splenial. Approximately 59 mm of the anterodorsal process of the right splenial was broken off when the element was found (Fig. 13). There is damage to the dorsal margin of the broken part and the dorsal margin of the splenial as a whole. Restoring this broken process and then placing the whole element in position on the jaw (using Brochu, 2003, Fig. 40 as a guide) this break aligns very well with the indented bone fragment in the dentary (ii) and the associated bite marks (i). This provides evidence that the right mandible was originally complete and one bite (or more, but very close to each other in position and delivery) damaged both the right dentary, and into and through, the posterior part of the right splenial (Fig. 14).

### Indeterminate

There are a number of pieces of the skull that present evidence of damage or breakage, but in which it cannot easily be determined if this was damage from trampling, bites, or general disarticulation/transport, etc. Therefore, although these may represent additional evidence for postmortem damage from another animal, we take a conservative approach here and do not directly assign them as such.

1. There is some form of scrape-like mark down the dorsal part of the left maxilla, close to the suture with the left nasal.

2. The left maxilla shows numerous breaks and damage. The base of the left maxilla (4.5 cm posterior to the anterior edge of the antorbital fenestra) is broken as is the posterioventral part of the maxilla, and some pieces of this element have separated from the main body (Fig. 2). These breaks can be aligned with the damage seen to the base of the left lacrimal, which is missing the anterior process and most of the ventral process, as well as the broken palatal shelf. All three areas (maxilla, lacrimal and palatal shelf) show evidence of breaks of green bone, and not just disarticulation or recent erosion. It is at least possible that this area was collectively damaged and/or removed as a result of a single or multiple bites to the area. This is plausible given the clear bite marks to the left dentary that shows postmortem bites to the skull, but the lack of distinct tooth marks on this part of the skull make it hard to confirm. This may simply be the result of damage through trampling, but then it would be difficult for an animal to step on a skull and break only one side, or leave the palate largely intact while the maxilla was shattered.

3. The anterior part of left pterygoid is broken and missing. The break occurs in a position approximately in line with the midpoint of the maxilla-jugal suture (based on comparisons

with other tyrannosaur skulls) and may be linked to the above described missing portion of the right maxilla and palate (point 2).

4. The anterior ramus of the right palatine is broken. Approximately 20 mm of bone missing based on comparison to the complete left element and the remainder of the anterior ramus is 67.5 mm long. This was found loose in the quarry, and was later restored to the main body of the right palatine. The posterior parts of both posterior rami of the right palatine are also broken and lost.

5. Left exoccipital has been broken away from the braincase. The braincase is relatively complete, but the left exoccipital suffered damage at its base and three loose pieces of this were found in the quarry. Two of these pieces can be fitted to the main body with the respective broken edges lining up well. There are no traces of bite marks or punctures suggesting that this was a natural break or caused through trampling.

6. A number of marks on the dentary teeth and isolated crowns may be pre- or post-mortem. These are not considered further as they may be as a result of combat, postmortem damage or malocclusion (Fig. 15). Two disarticulated teeth (one already broken) show possible marks that are similar to tooth marks on tyrannosaurine teeth held in the Tyrrell collections (e.g., TMP 1988.050.0145, TMP 1988.050.0146) but it is not possible to determine when these marks occurred.

## Position of elements in field

The orientation and position of a number of elements is important to some of the interpretation of the data, and so is described here. Figure 16 is a redrawn version of the quarry map by DHT (who also helped excavate the material and was one of the main preparators of the specimen, and prepared the skull), which was originally made by the excavation team and provides reference for this description.

The intact mandible was found with its ventral edge uppermost lying over the intact palate with the cranium being inverted and the ventral side uppermost. The premaxillary teeth of the cranium were interlocked with the fourth dentary tooth (approximately—see Fig. 17) and the first three dentary teeth were not *in situ* in the mandible, but instead were found inside the palate. These were recovered in the premaxillary-nasal area of the skull, and the missing posterior dentary teeth were located deep in the skull cavity. Collectively this shows that the original position of the dentary was close to natural articulation with the rest of the skull in order for the teeth to fall out and come to lie within the depths of the skull. This also shows that the head could not have been encased in sediment at this time in order for the teeth to have travelled so far down into the cranium. We suggest that the dentary was therefore only later moved up and anteriorly, leading to the near separation of the teeth in the dentary and their posteriorly directed alignment—a pattern not seen in the teeth of the premaxilla and maxilla.

The femur was buried when the specimen was first discovered, therefore the damage it has suffered (major breaks and crushing) is considered likely to be through trampling and thus Cretaceous in origin. The ilium however was eroded and the damage to this is recent.

## MATERIALS AND METHODS

Impressions of the positions and spacing of premaxillary and anterior maxillary teeth and the anterior dentary teeth of a number of Dinosaur Park Formation theropod skull casts were taken in order to examine the arrangement of their teeth. This allowed comparison with the spacing of tooth marks on the *Daspletosaurus* elements. Pieces of 50 mm thick conservation-grade Ethafoam were pressed lightly onto casts of tooth rows and the spaces between the impressions of the teeth recorded.

## DISCUSSION

### Premortem injury

Studies suggest cranio-facial biting may have been common in tyrannosaurs and possibly other large theropods (*Tanke & Currie, 1998*; *Peterson et al., 2009*; *Bell & Currie, 2010*). Some extant large carnivores will engage in fights to the death (intra- and interspecifically), either over territory/food or one actively killing another for food. However, such engagements are rare, if only because of the typically low numbers of large carnivores in an ecosystem. In addition, animals also try to avoid injury and another large, adult, healthy carnivore is clearly equipped to potentially inflict serious or lethal injuries. This is in contrast to more typical prey of small juveniles that also typically lack adult defences such as well-developed horns (*Hone & Rauhut, 2010*). Currently, evidence of non-avian theropods consuming others (even if there was apparently a large size difference between the consumer and consumed—*Sinocalliopteryx*—*Xing et al., 2012*) is uncommon, and while these would be expected to be rare, they are clearly occurring. Carnivorous theropods clearly interacted with other live carnivorous theropods, in at least some cases likely hunting and killing them, and consuming dead ones.

Injuries observed on dinosaurs that are hypothesized to be the result of attempted predation by a theropod (e.g., *Murphy, Carpenter & Trexler, 2013*; *DePalma et al., 2013*) are typically on the body of the intended prey and especially towards the rear of the animal where one might expect a pursuing predator to strike. Although feeding traces from theropods have been found on the cranium of a consumed dinosaur (*Hone et al., 2010*) this was notably a situation considered late-stage carcass consumption with bites being normally expected to occur first in the areas of main muscle mass (e.g., the muscles of the pelvis and hindlimbs, viscera etc.—see *Hone & Watabe, 2010*).

In contrast, injuries hypothesized to be the result of intraspecific biting are localised on the cranium, implying the animals faced each other directly, or lined up side by side and thus potentially in some form of ritualised combat. In the case of the specimen here, despite the lack of much of the postcrania and the injured dorsal rib, it is clear that numerous injuries were inflicted on the skull at some point (and perhaps on multiple occasions) some considerable time before death (see *Straight et al., 2009*). Based on the size and scale of the injury, at least some of these would appear to have been inflicted by another large theropod. Their form—punctures, lesions and scrapes are consistent with large and subcircular teeth and the pattern and positioning seen here—is similar to that on another bitten tyrannosaurine skull (*Peterson et al., 2009*).

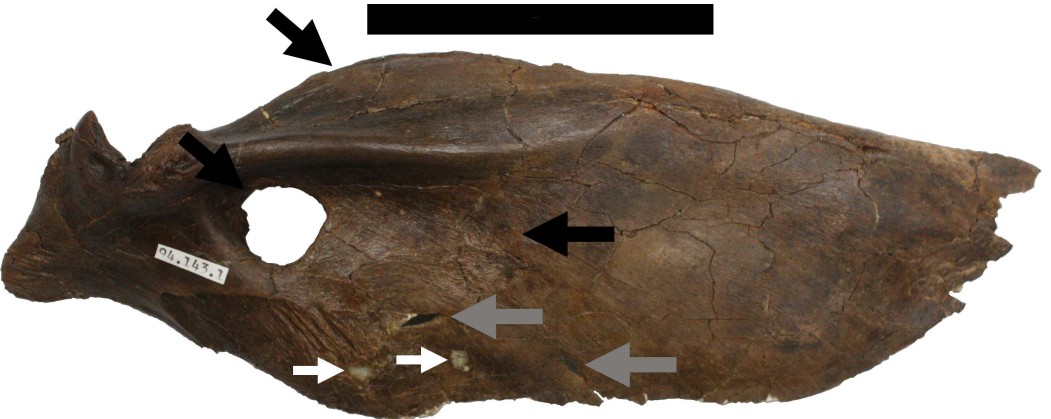

**Figure 9 Damage to right surangular.** Damage M to the right surangular. Several areas showing osteomyelitic bone are indicated with black arrows. Two holes that appear to be a result of damage or infection are indicated with grey arrows, and two holes that are preparation marks are indicated with smaller white arrows. Scale bar is 100 mm.

The most obviously candidate to have inflicted these marks is another tyrannosaurid since these are the only large theropods known in the Dinosaur Park Formation and the damage to the back of the head includes a puncture wound attributable to a large, sub-circular tooth (K— Fig. 7) that is characteristic of the group (*Holtz, 2004*). Given that such ritualised combat would appear to be more likely between two conspecifics than heterospecifics, we conclude that this was likely through engagement with another *Daspletosaurus*. At least two other non-adult tyrannosaurids from the Late Cretaceous of North America show evidence of cranio-facial biting (*Tanke & Currie, 1998*; *Peterson et al., 2009*) suggesting such engagements were not limited to mature animals.

One additional aspect of the specimen supports this general interpretation. The two circular holes seen in the surangular (Fig. 9) are similar in form to a number of injuries in various tyrannosaurids that are refereed to trichomonosis-type infection (*Wolff et al., 2009*). These have been identified on several surangulars of various tyrannosaurid taxa (*Wolff et al., 2009*, their Fig. 2) and these and similar lesions were inferred to be the result of infections transmitted by intraspecific aggression.

It is possible that some of the premortem damage to the cranium was inflicted by the pes of a *Daspletosaurus*. *Rothschild (2013)* recently noted that some injuries to a *Tyrannosaurus* skull may have been inflicted by the feet of a conspecific, and if this is the case, it is reasonable to assume other large-bodied tyrannosaurids may have fought in a similar way. However, this is considered an unlikely possibility given the obvious difficulty of large bipeds using the feet to attack the cranium of another animal. In the case of the damage to the occipital region at least (K) this would appear to be a bite, and we suggest that that in combat between two tyrannosaurines injuries to the head are generally more likely to be inflicted by the jaws than a pes.

Only a limited number of the injuries described here can be ascribed to intra- (or even inter-) specific combat. In addition to the possible results of infection seen in the surangular, a number of areas of damage are slight or do not obviously match a bite

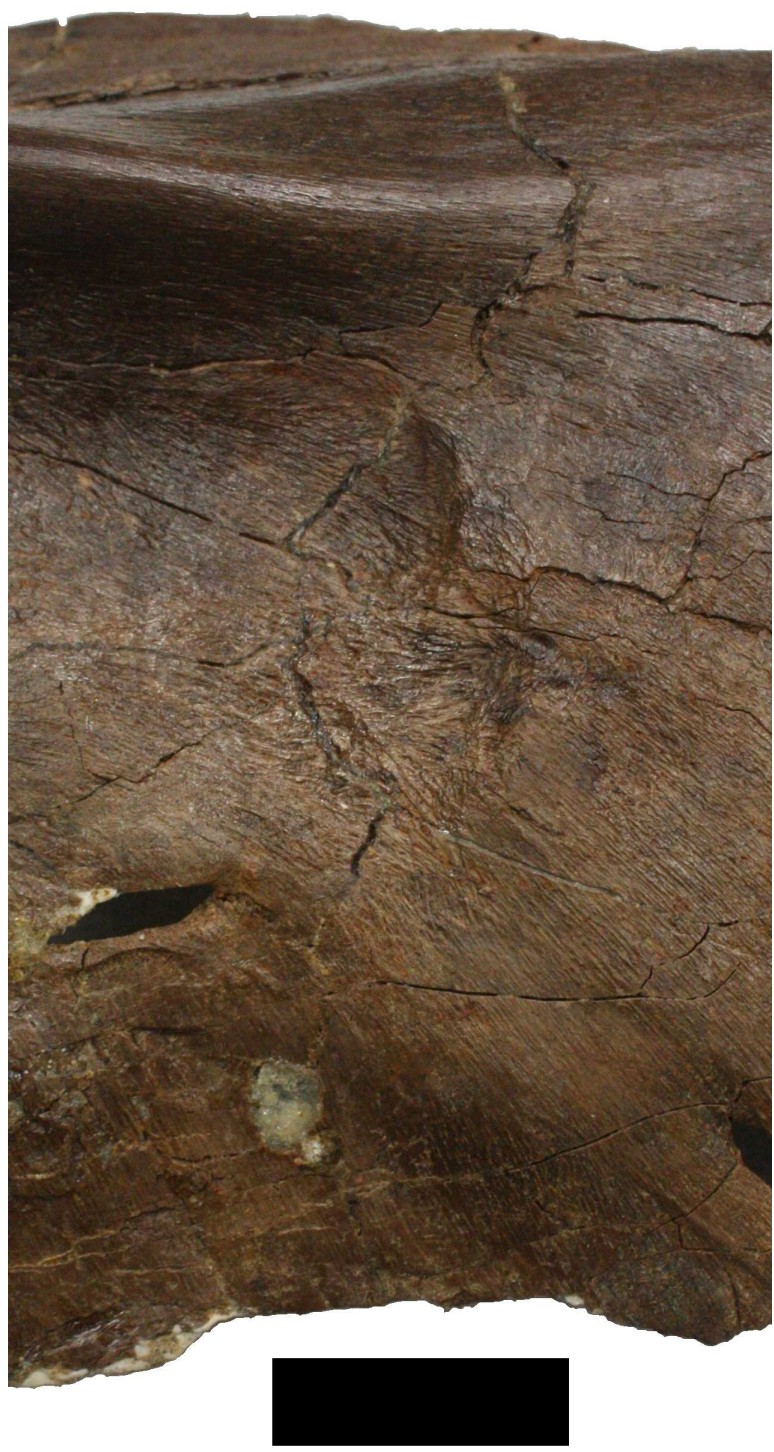

**Figure 10 Close up of osteomyeltic bone.** Close up of the major patch of osteomyeltic bone on the right surangular (central part of the image). Scale bar is 20 mm.

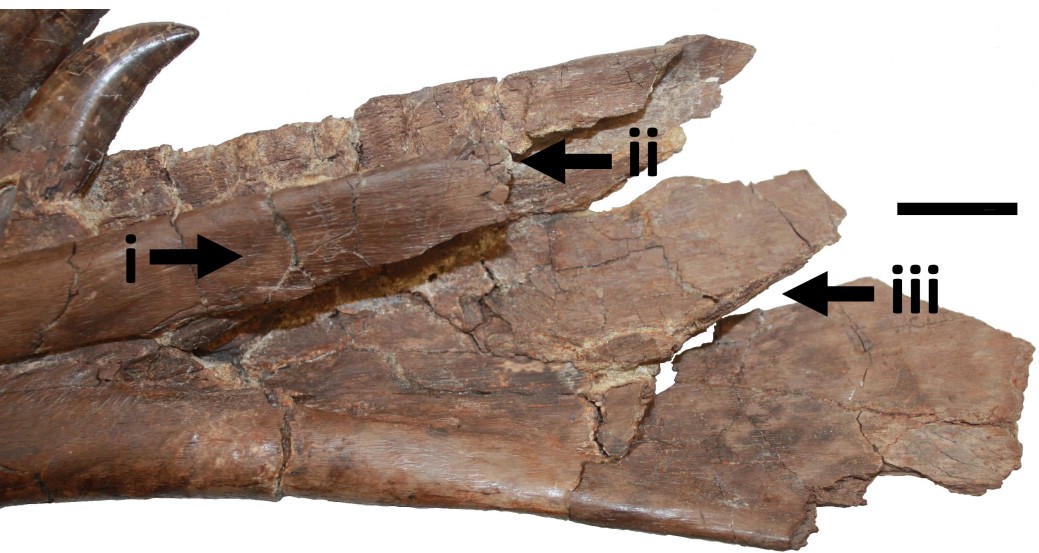

**Figure 11 Damage to posterior part of dentary.** Damage i, ii and iii to the posterior medial part of the right dentary. A series of vertically aligned scores (i) and a break in the bone (ii), and damage to the lower part of the dentary (iii) likely as a result of postmortem bites. This shows the element as currently preserved where a fragment of bone is missing posterior to ii—see also Fig. 12. Scale bar is 20 mm.

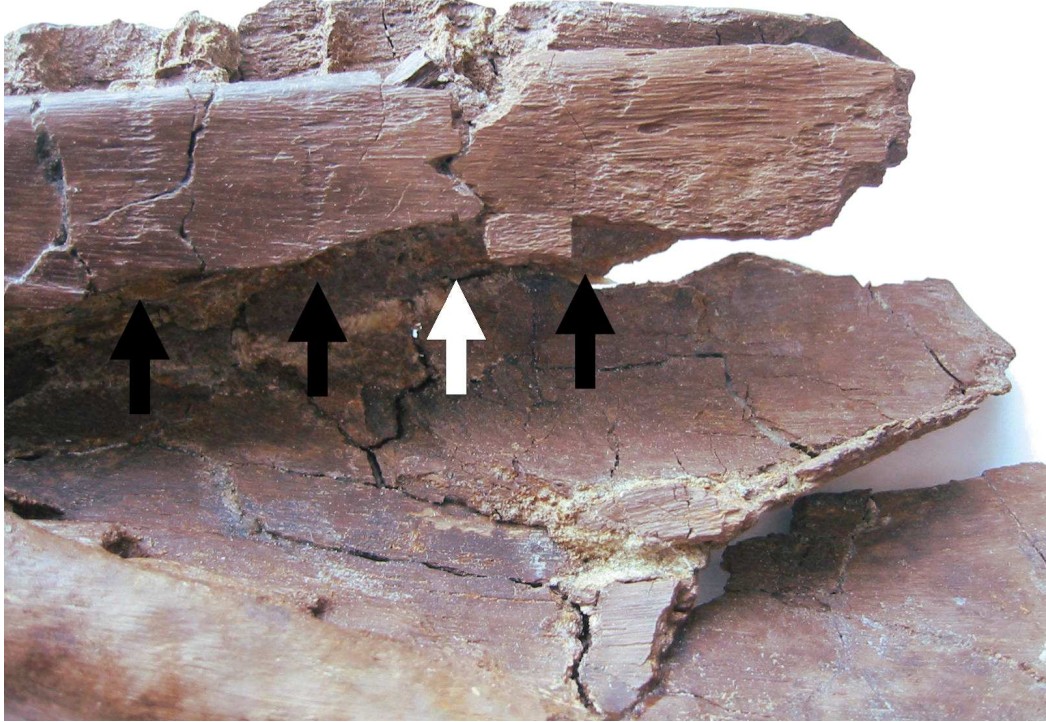

**Figure 12 Detail of dentary damage.** Damage i and ii as seen prior to the loss of the part of the dentary to the right of the break from damage ii (white arrow). The bite marks on the medial surface of i (black arrows) are also clearly seen.

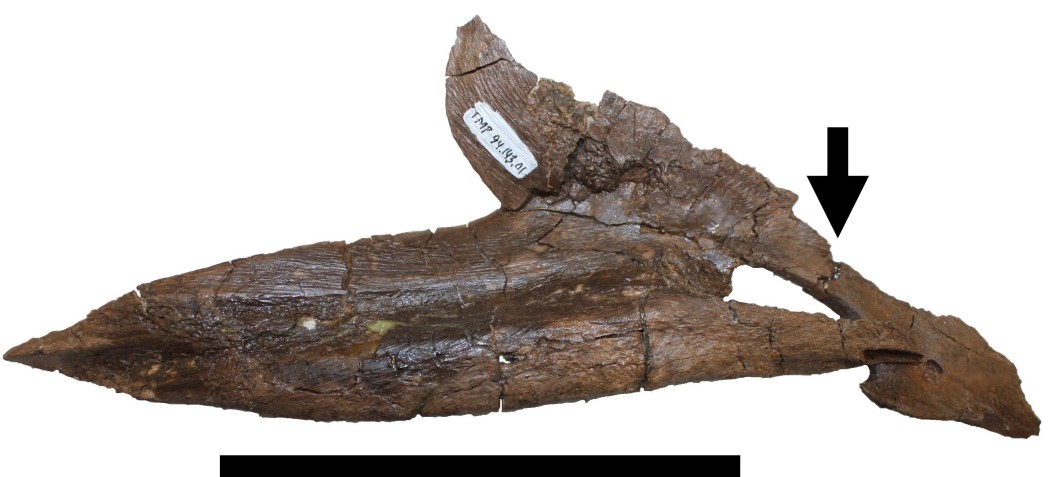

**Figure 13 Damage to right splenial.** Right splenial in medial view showing damage iv. The part to the right of the black arrow indicates the piece that was separated. Scale bar is 100 mm.

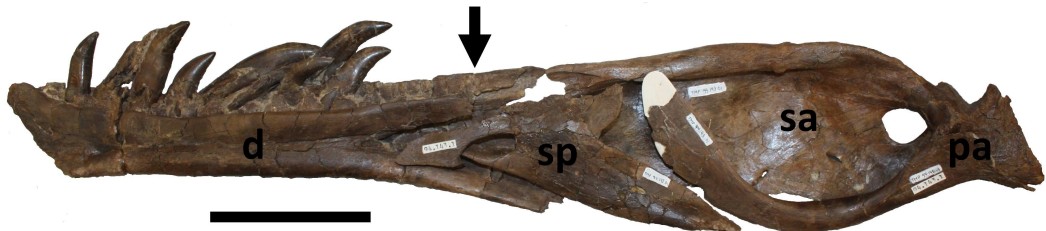

**Figure 14 Reconstruction of right mandible in medial view.** Approximate reconstruction of the elements of the right mandible in medial view. Note that the black arrow indicates the damage ii inflicted on the dentary (see Fig. 11) is closely aligned to the break on the anterior part of the splenial (Fig. 13), suggesting the two breaks may have occurred as the result of a single bite. Abbreviations: d, dentary; sp, splenial; sa, surangular; pa, prearticular. Scale bar is 100 mm.

pattern. For example, the patches of osteomyeletic bone near the promaxillary fenestra and the surangular (Fig. 9) and other small areas could be the result of any small infection or injury. There is greater confidence with regards to the puncture-type damage to the snout (Fig. 3), deep score on the maxilla (Fig. 5) and the damage to the occipital region (Fig. 7—that also appears to include a tooth-shaped puncture wound) such that at least some injuries can be strongly associated with apparent intraspecific combat.

### Postmortem bites

The postmortem bites inflicted on the skull also indicate the bite-maker was a large tyrannosaurid as other large carnivorous tetrapods from the Dinosaur Park Formation can be ruled out. The spacing between the marks of the teeth imply a large animal and the possibility that a large part of the skull was bitten through also rules out smaller theropods and small carnivores such as champsosaurs. Azhdarchid pterosaurs may have scavenged on dead animals, but these animals are edentulous and would not have left tooth marks. A number of crocodilians are known from the formation, but these are small (skull length

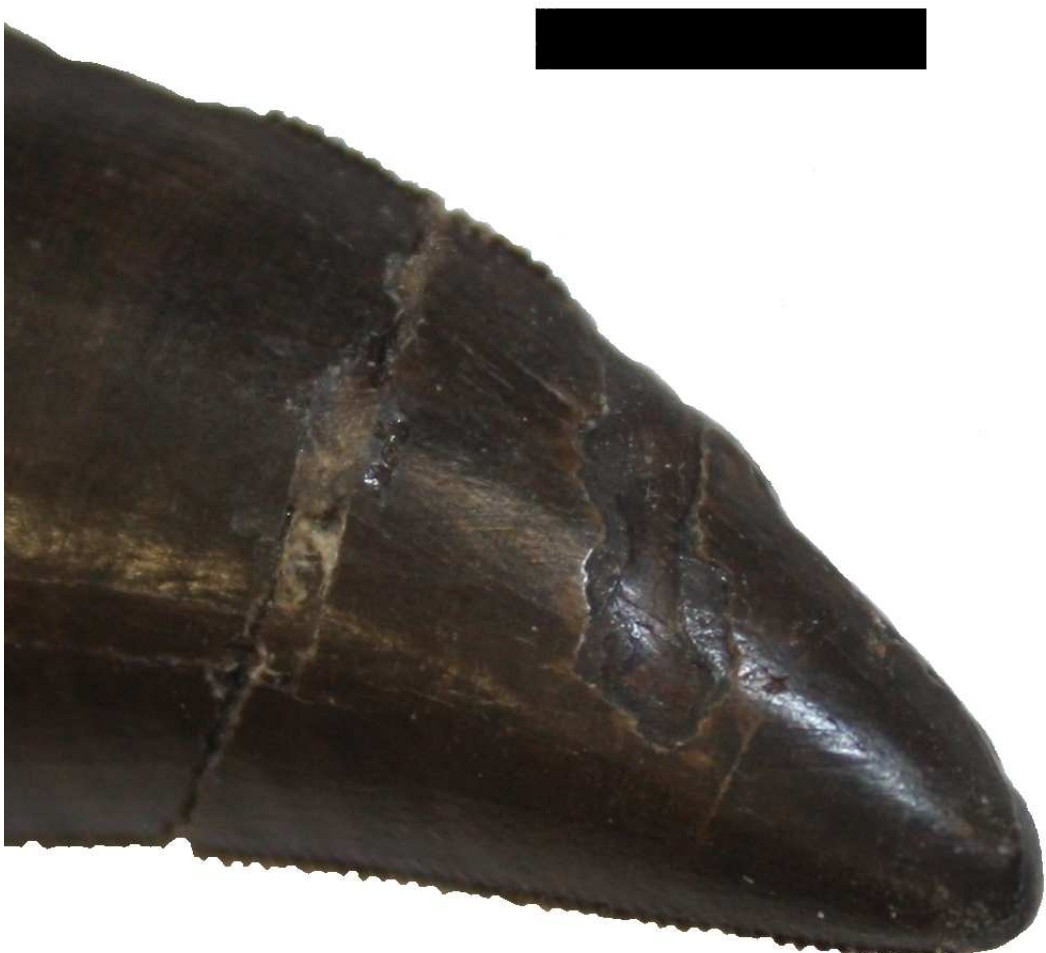

**Figure 15 Wear on a dentary tooth.** Isolated dentary tooth showing occlusal wear on the enamel (damage 6). Scale bar is 10 mm.

under 50 cm—*Wu, 2005*) and thus unlikely to have left such large marks. In addition, as modern forms at least feed primarily through torsion, these would not be expected to leave such straight bite marks on the tyrannosaur mandible.

This immediately suggests that the marks were made by a tyrannosaurid, and they do at least bear some resemblance to feeding traces made by tyrannosaurines (e.g., *Erickson et al., 1996*; *Hone & Watabe, 2010*). Examination of tooth spacing for bite marks is most inexact—although a large space between tooth marks is typically indicative of a large animal, incompletely erupted teeth, or missing/damaged teeth in a jaw can lead to large spaces between tooth crowns that contact bone. Combined with the range of sizes of teeth in the jaw of a single theropod, the difference in spacing between maxillary and premaxillary teeth, the differences in sizes between individuals and intraspecific variation, it may be impossible to tell one species from another from bite marks alone. However, it may be possible to at least rule out some candidates and/or determine if bites were made

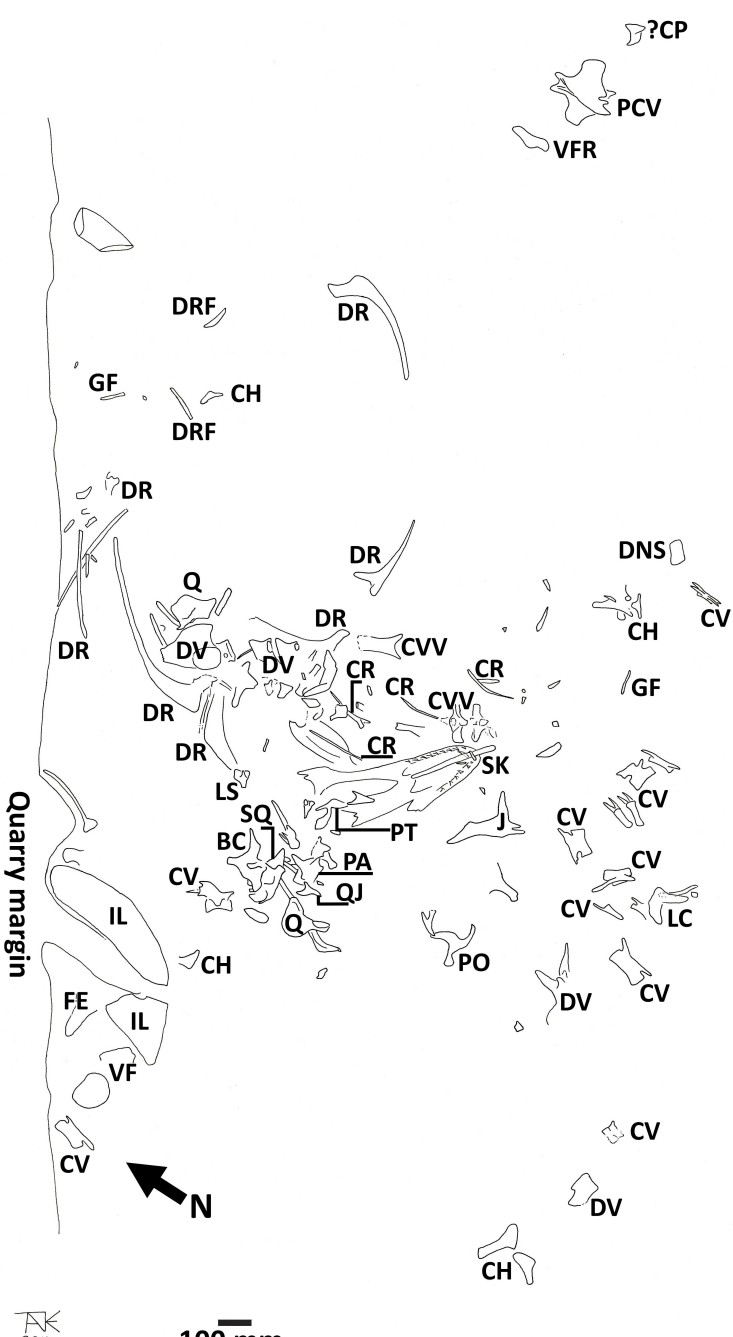

**Figure 16 Quarry map of the *Daspletosaurus*.** Key to elements is as follows: BC, braincase; CH, chevron; ?CP, coronoid process; CR, cervical rib; CV, caudal vertebra; CVV, cervical vertebra; DNS, dorsal neural spine; GF, gastralium fragment; DR, dorsal rib; DRF, dorsal rib fragment; DV, dorsal vertebra; FE, partial femur; Il, ilium; J, jugal; LC, lacrimal; LS, laterosphenoid; PA, prearticular; PCV, proximal caudal vertebra; PO, postorbital; PT, pterygoid; Q, quadrate; QJ, quadratojugal; SK, skull (major piece including a dentary); VF, vertebral fragment. Fragmentary elements that can be identified as belonging to the skeleton are illustrated but not labeled, and elements from other taxa are not shown. North is indicated on the map, and the scale bar is 100 mm.

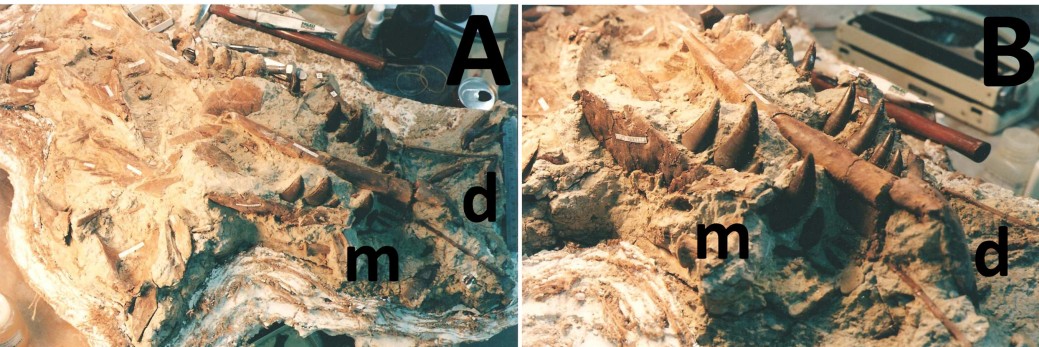

**Figure 17 Position of the mandible during preparation.** The specimen under preparation showing the position of the right dentary relative the main part of the skull (see also Fig. 16). (A) in laterodorsal view with the maxilla (m) and dentary (d) visible, and (B) in anterodorsal view. The dentary is offset from its natural position but also moved considerably anteriorly.

using maxillary teeth or a premaxillary arcade, especially when there is heterodonty as with the tyrannosauids.

The position and spacing of the teeth of various Dinosaur Park Formation theropods was assessed using foam impressions (see above). However, given the variation seen within taxa (and especially given the different sizes of even single species) it is difficult to make any firm assessments. Even so, both the size of, and the spacing between, the marks recorded would appear to rule out small bodied theropods. It is certainly plausible that the mark-maker was a second *Daspletosaurus* and there is nothing to suggest that it was not. As above, given the limited number of tyrannosaurs in the Dinosaur Park Formation, *Daspletosaurus* remains a strong candidate but there is too little information to separate out this from contemporaneous tyrannosaurids. Cannibalism is recorded in *Tyrannosaurus* (*Longrich et al., 2010*), and it is likely that other theropods, including other tyrannosaurids, fed on each other.

The one definitive bite on the *Daspletosaurus* dentary is likely the result of a bite involving the maxillary or non-anteriormost dentary teeth. The premaxillary arcade and/or anterior dentary teeth would be much more closely appressed to one another, even in the largest of tyrannosaurines. This would result in tooth marks that were much closer together, hence the suggestion that these are from a more lateral part of the tooth row. Strong bites from tyrannosaurs seem to be typically delivered with the anterior part of the tooth arcade (*Erickson et al., 1996*) but these are also used for scrape-feeding to remove flesh from bones (see *Hone & Watabe, 2010*) which contrasts with the pattern seen here. This may be accidental tooth-on-bone contact therefore, or the result of a bite directed at another part of the skull that left these incidental marks here. Given the apparent delicacy and selective feeding strategies employed by at least one tyrannosaurine, this mark may potentially represent a deliberate feeding strategy, scraping the maxillary teeth alongside the dentary in an attempt to remove muscle tissue, but additional specimens are needed to confirm this hypothesis.

### Taphonomy/burial and evidence for scavenging

Collectively the specimen most closely resembles Taphonomic Mode B of *Eberth & Currie (2005)* for the Dinosaur Park Formation. That is, although the specimen is largely incomplete the vast majority of the recovered material relates to a single individual skeleton. These are primarily recovered in palaeochannel facies (*Eberth & Currie, 2005*) and some common themes in such theropod specimens (their Table 24.2) suggest the possibility of local reworking, and that the animal may have died through drowning or disease/injury. Easily transported microvertebrate fossils were found scattered in the matrix that surrounded the tyrannosaur suggesting some fluvial action. The preserved microvertebrate specimens do not reveal anything unusual with respect to the usual Dinosaur Provincial Park fauna but the presence of champsosaur, crocodylian, and trionychid material among others does also indicate potential fluvial conditions. However, the presence and preservation of the small bone fragments that could be identified and restored to various parts of the main skull and dentary especially show that the water movement was slow.

The specimen as recovered was somewhat dissociated and it is difficult to separate out which of the disarticulation and damage to the skeletal elements may be the result of feeding, and which from simply collapse of a decomposing carcass or possible transport. Certainly the specimen suffered some damage from another theropod, and the damage to the femur suggests damage through trampling. However, the lack of much of the postcranium and the presence of elements from other species in the quarry points to a degree of transport in the local area that moved elements around at times.

For example, the position of the left jugal and lack of damage compared to the left lacrimal may be as a result of the dissociation of the former, followed by a bite or trampling damaging to the latter, but could also have resulted from a bite damaging the lacrimal with the jugal being separated as a result and falling away (Fig. 2). Thus, both the positions of marks and breaks on the individual elements (and lost parts or broken rami found elsewhere in the quarry) must be assessed against their likely original positions. Furthermore, a heavy bite on thin bones (as with a cranium) would not leave deep tooth impressions (cf. *Erickson et al., 1996*; *Hone & Watabe, 2010*) but instead might lead to shattering, or at least breaking, of elements. Thus, tooth marks might not be left or easy to discern, despite a strong bite. Our inferences below are therefore tentative given that some major breaks appear in patterns that imply breakage of multiple aligned elements, but without the definitive evidence that a tooth mark would provide.

There is a lack of bite marks on the remaining parts of postcranial skeleton, including areas where meat would be plentiful on a fresh carcass, or even mid-stage carcass consumption (following *Blumenshine, 1987*) e.g., the ilium, femur, ribs. However, tyrannosaurids seem to leave far more bites on material that do other carnivorous theropods, and even may leave multiple traces with relatively 'careful' feeding (*Hone & Watabe, 2010*). However, given the amount of missing material it is likely that the specimen was exposed for some time prior to burial. The lack of bite marks may therefore be the result of the loss of much material which could have borne them, although and the lack of

shed teeth (both of tyrannosaurs and other small carnivores e.g., see *Hone et al., 2010*) also collectively suggest that the material was not fed upon extensively.

If the animal was relatively well-fleshed the when the carnivore fed upon it, then it would be possible to consume large amounts of material without breaking bones or leaving tooth marks. However, the lack of damage and separation of gastralia and other fine elements like the supradentary suggests this was not part of a normal carcass consumption pattern. The cranium had clearly disintegrated at least in part and elements had separated along suture lines prior to burial, and not primarily as a result of being bitten.

The cranium at least must have been undergoing some decay when the bite occurred. This is based on the extrusion of the dentary teeth, which must have been held in by their ligaments in order not to have fallen out entirely (Fig. 8), but still loosely enough attached that they were all partly extruded from their sockets. Similarly, the loose teeth that had fallen into the palate must have decayed to the point that the ligaments were no longer sufficient to hold them in place. We cannot easily estimate how long it may have taken for tyrannosaur tooth ligaments to decay and separate, but presumably this would have been minimally a number of days rather than hours. Therefore the action that lifted the dentary and repositioned it, presumably coincident with the marks delivered to the dentary, occurred some days after death.

Based on the orientation of the dentary teeth and the preserved position of the dentary, this must have been moved anteriorly (Fig. 17). Given that it was originally in a natural articulation (based on the dentary teeth lying in the palate), it must have been lifted to come into its final resting place. It could not simply have been moved forwards by fluvial action as the teeth of the dentary and maxilla would have interlocked. The delivery of the bite that broke this part of the skull is also very unlikely to have been during intraspecific combat or a predation event given the position of the bite marks up inside the jaws and in such a posterior location. The *Daspletosaurus* would need to have opened the mouth to an extraordinary degree to allow the jaws of the other animal to have reached this position. A decaying carcass however would likely present no such difficulties.

As described above, two other areas of damage are hard to confirm as bites as opposed to damage through trampling. In addition to the damage to, and movement of, the left dentary, there is a possible bite into the right jaw, damaging the right dentary, right splenial and perhaps the right pterygoid. A possible second bite would have been into the left side of the face, damaging the left lacrimal and left maxilla. The left jugal was displaced and was recovered close to the main part of the skull (see Fig. 16). If this element has dissociated through decomposition prior to the putative bite it would explain why this is undamaged, despite the major trauma to that side of the face.

There may also have been a bite to the missing left dentary. Based on *Daspletosaurus* having four teeth in each premaxilla, 17 for each maxilla (*Currie, 2003*, though he notes this specimen has an unusually low maxillary count and it may be as little as 13) and 18 for each dentary, the total number of teeth in the skull would have been up to 74. A total of 31 erupted teeth are present *in situ* in the skull and dentary (not including the incipient replacement teeth), and 19 loose teeth were collected from the quarry, a total of 50 teeth.

These teeth match those in the cranium for size, shape and colour (and retain roots) and are not considered shed teeth by another animal. Therefore, we conclude that originally the left dentary must have been present in the immediate area and shed nearly all of its teeth at some point before it was lost.

Finally, the skull is the tallest element of the specimen, and even allowing for stratification of some pieces of the postcranium, may have been the uppermost material in the quarry. Thus, assuming there was a low level of sediment/water in the environment, the skull may have been the only exposed piece, or the most exposed piece, when the scavenging took place. This is somewhat speculative, but fits the overall pattern of marks on the skull not seen on any other material and the apparent trampling of a buried femur. This may therefore explain why one, or even both, dentaries were bitten—they were simply the only exposed part of the skeleton when the encounter took place and thus did not follow the expected carcass consumption patterns.

As with the premortem injuries, it is not possible to distinguish easily between cannibalism and feeding by another tyrannosaurid. *Currie (2005)* notes that *Gorgosaurus* is generally more common in the Dinosaur Park Formation, and thus based simply on numbers would appear to be a more likely candidate to have fed on this animal than a second *Daspletosaurus*. Cannibalism is known in North American tyrannosaurines (*Longrich et al., 2010*) and thus should not be ruled out in *Daspletosaurus*. Certainly it seems most likely that a tyrannosaurid, if not necessarily a conspecific, was responsible for the postmortem feeding traces left here.

## CONCLUSIONS

In summary, this *Daspletosaurus* skull suffered numerous injuries to both the cranium and mandible that were both pre- and postmortem. Numerous wounds were inflicted during life and despite some considerable damage (especially to the occipital region) the animal clearly survived as shown by the evidence of healing. These were likely inflicted at least in part by one or more conspecifics and perhaps as a result of numerous separate instances of conflict. After death, the animal suffered at least one major bite (to the right dentary), and perhaps two more (a bite to the missing dentary, and finally one to the left maxilla and associated areas) from another tyrannosaurid, possibly another *Daspletosaurus*. The specimen must have been decaying prior to the delivery of the bite to the dentary, and the condition of other material suggests scavenging, rather than simply late-stage carcass consumption. These interpretations must remain tentative, but this is considered a possible scenario of cannibalistic scavenging and is strong evidence for one large tyrannosaur consuming another, in addition to providing further support for cranio-facial biting in tyrannosaurs.

**Institutional Abbreviation**

TMP          Royal Tyrrell Museum of Palaeontology, Alberta, Canada.

## ACKNOWLEDGEMENTS

We thank the following for access to material in the Tyrrell collections: Brandon Strilisky, Graeme Housego, and Tom Courtenay. Rhian Russell is thanked for her work repairing some breaks and damaged parts of the material which allowed better examination of the specimen and of course those who originally collected and prepared this remarkable specimen. Thanks to Patty Ralrick for scanning the photographs that were used to compile Fig. 17 and to Malarie Jane Shaffer for assistance with the quarry map. Don Henderson is also thanked for his hospitality and discussions of tyrannosaur jaws and Jenny Clack for her editorial work on the manuscript and the comments of the referees Thomas Holtz, Steven Brusatte and Joseph Peterson who contributed to its improvement. Finally we thank Experiment.com and the contributors to this project for their support with this project.

### Funding

The work was supported by donations via Experiment.com and contributions of numerous donors including Adrian Allen, Marko Mosscher, Brendan Clarke, Herman Diaz, Daniel Grimes, Keith Guerin, Joseph R. Hancock, Kilian Hekhuis, Tom Holland, Oliver Humpage, Megan, Harry and William Robbins, and other anonymous contributors. Special thanks in particular to Denny Luan, David Orr, Matt van Rooijen, Luis Rey, and Brett Booth for their donations and support. The funders had no role in study design, data collection and analysis, decision to publish, or preparation of the manuscript.

### Competing Interests

DH Tanke is an employee of the Royal Tyrrell Museum of Palaeontology.

### Author Contributions

- DWE Hone and DH Tanke conceived and designed the experiments, performed the experiments, analyzed the data, contributed reagents/materials/analysis tools, wrote the paper, prepared figures and/or tables, reviewed drafts of the paper.

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
