# Peer review of "Pre- and postmortem tyrannosaurid bite marks on the remains of Daspletosaurus (Tyrannosaurinae: Theropoda) from Dinosaur Provincial Park, Alberta, Canada"

_PeerJ, doi:10.7717/peerj.885_

## Round 0.1 · original submission · Major Revisions

Although the template text refers to this decision as a ‘major revision’ I wanted to explain that it has not yet been sent for review. Before sending the submission out to reviewers, I read this submission fully myself and in my opinion there are several issues with the spelling, terminology and grammar which should be fixed before it is sent out to review (in order to ensure the best review experience). Please address the changes in my annotated manuscript, which the Editorial Office will send you separately, and re-submit.

---

## Round 0.2 · Minor Revisions

Dear David, here are the three reviews obtained for your paper. Two of them suggest very minor changes, one, more substantial ones. You will see that some of the suggestions correspond to ones I made earlier, including some typos. Please try and condense some of the text, if possible. I note that there is another paper on bite marks has just appeared in PeerJ (https://peerj.com/preprints/824v1/). Do you want to cite it? I also want you to explain the term 'foam' more precisely. What sort? Give a commercial name if it has one. How applied? Did you make a permanent record from the results? (Should go in M&M) Jenny

·

Basic reporting

See general comments

Experimental design

See general comments

Validity of the findings

See general comments

Additional comments

I read this manuscript on tyrannosaurid pathologies and feeding habits with interest. The specimen the authors are describing is a very intriguing fossil, with a variety of injuries that deserve full description, and in doing so, may reveal some new things about how tyrannosaurids fed, interacted with each other, and dealt with injuries. I think this paper should be published, but I recommend some revisions before acceptance.

I will also state that I am not an expert on pathologies, injuries, or feeding traces, so I cannot judge the very specific descriptions and diagnoses made by the authors (in terms of identifying individual injuries as pre or post-mortem based on bone texture and signs of healing, in terms of identifying specific bone injuries such as osteomyelitis, etc.). With that said, I think the rationale the authors present when describing these injuries is clear, and makes logical sense to an outsider.

I have a few major points that the authors should focus on during their revision, and several more minor points. I anticipate that the authors should be able to address all of these issues without too much trouble, and at that stage the manuscript should be publishable. I look forward to seeing this gorgeous specimen in print!

Steve Brusatte, Univ of Edinburgh, February 3, 2015

Major points:

Identification of the specimen: The authors need to beef up the evidence for referring this specimen to Daspletosaurus. They state that “this specimen can be confidently assigned to this genus”, but I don’t think they have demonstrated this clearly. They are probably correct in identifying it as Daspletosaurus, but Daspletosaurus does comprise several species with quite a lot of variation, and it does overlap with other tyrannosaurids in time and space (particularly Gorgosaurus), and in the past Daspletosaurus specimens have been mistaken for those of other tyrannosaurids (and vice versa). The rationale for assignment to Daspletosaurus is a bit woolly. Ideally the authors should note autapomorphic characters that are present in the specimen, although I realize that Daspletosaurus is not a very well described tyrannosaurid in the literature (although this will change with the in-progress revision by Thomas Carr, which has been presented in some pieces at SVP meetings). I suggest the authors do two things: First, talk to Thomas Carr and see what he thinks, and whether the specimen does exhibit any clear Daspletosaurus characters that you can cite as a pers comm, or perhaps those he has discussed in his SVP abstracts. Second, score the specimen for the phylogenetic dataset of Brusatte et al. (2010) and see where it falls out. The authors do mention that the specimen has one character that “is diagnostic for (Daspletosaurus) compared to other tyrannosaurines” based on the Brusatte et al. dataset, but this is very unsatisfactory, because it is only one character, and this rationale assumes that the specimen is a tyrannosaurine (something the authors haven’t demonstrated. Could it be an albertosaurine?). Also, be sure to double check the tooth count data and the phylogenetic utility of this feature. As the authors are surely aware, tooth counts are highly variable in tyrannosaurids, both individually and also ontogenetically. We discussed this in the 2012 Brusatte et al. Alioramus monograph and I would refer the authors to this.

Identification of the specimen as a juvenile: Based on the size of the specimen, the authors are probably correct that it is a juvenile. However, we now know that there were tyrannosaurids with fairly small adult size in the Late Cretaceous of Alaska, and maybe also in Asia. So size alone is not a good proxy. The authors vaguely mention that some bones are unfused. Which bones? And after identifying these bones, the authors need to demonstrate that their fusion (or lack thereof) is a good correlate with maturity. There has been a lot of literature on this topic. There has also been a vast literature on how discrete characters change during tyrannosaurid ontogeny (primarily the work by Thomas Carr). This should be consulted and the authors should make clear which fusion and discrete characters support a juvenile status.

Taphonomy: Throughout the manuscript, particularly in the Preservation of Material section, the taphonomy of the specimen and the larger quarry are discussed fairly vaguely, and qualitatively. The authors say things like “the overall condition of the material suggests that it was buried in situ in the fine-grained silt” and “there is no evidence of fluvial action.” All of this needs to be supported more rigorously, and ideally more quantitatively. The authors could apply various numerical taphonomic metrics, such as abrasion scales and Voorhies group analysis, to the specimen. This doesn’t have to be done to death; just a few analyses will suffice. It is important, though, because making sure you have the taphonomy right is important when identifying the various bumps, lesions, etc. and interpreting whether these are biological injuries or preservational/taphonomic artefacts. I suggest taking a look at a paper my colleagues and I wrote a few years ago on a Triceratops bonebed (Mathews et al. 2009, JVP). We performed some of these analyses and then used them to interpret how this collection of Triceratops formed. I’m not saying the paper should be cited (!!), but just that it is a brief example of some of the analyses that you can do, with references to the major studies.

Face-Biting: I am not quite convinced by the portion of the Discussion “Premortem Injury” in which the authors interpret many of the marks on the face as inflicted by another big theropod, and therefore evidence of theropod-on-theropod face biting for dominance/combat. There is other evidence for this kind of behavior in large theropods, so I think it is very plausible. However, I do not see clear evidence on this specimen, at least based on the descriptions and photos in the manuscript. The authors have identified a number of lesions/marks/scratches on the facial bones, but how many of these are clear bite marks? How many of these match (or could match) the teeth of tyrannosaurids? In this discussion, the authors mention only a single mark (pathology K) which seems to be a puncture that matches the size/shape of a large theropod tooth. Some of the other pathologies in the Description are referred to as bites or described as possibly being related to bites, but these are not reviewed and summed up in the Discussion. I suggest the authors take a step back and state explicitly in the Discussion which of these facial marks are plausibly related to bites, what makes them bites and not another type of injury, and of these bites which are of the size/shape/orientation to be matched to other large theropods. Additionally the authors state that some of the features in their skull “are consistent with large and subcircular teeth and the pattern and positioning here is similar to that on another bitten tyrannosaurine skull”, citing a paper by Peterson et al. This needs to be expanded on: what specifically is similar between the new specimen and the specimens described by Peterson et al., which indicates that there is a similar pattern of facial biting? I’m pretty sure the authors will be able to provide good evidence for facial biting in their new specimen, but it’s just not presented very clearly in this version of the manuscript.

Small points:

There are still various spelling and grammatical mistakes in the ms, so the authors should again go through it carefully. For example, sagittal crest is the proper spelling (not saggital).

In the abstract, do not use “carnivorans” in the first sentence. Carnivorans properly are a group of mammals (dogs, cats, etc.). I think you mean carnivores.

In the introduction, when the authors say that tyrannosaurs “potentially (consumed) bones”, I think this can be said more definitively, because we know from the T. rex coprolite that at least one individual tyrannosaurid ingested a lot of bones, which is consistent with a lot of other evidence for bone ingestion that the authors mention.

In the Description, the authors cite a number of recent papers on tyrannosaurine cranial morphology, ontogeny, and taxonomy. Not to dig for citations, but our monograph of Alioramus should be added here (Brusatte et al. 2012, AMNH Bulletin) as it is probably the most detailed (or at least lengthy…) published description of a tyrannosaurine skull. The Bever et al. AMNH Bulletin on the braincase of Alioramus probably also should be mentioned.

In the Description, it is stated that the specimen has “alveoli for three premaxillary teeth”. Is this real or a typo? If real, does that mean one of the alveoli is broken off or were there genuinely only three alveoli in life? A tyrannosaurid with only three premaxillary teeth would be very strange indeed!

In the Description, of pathology K, aside from the misspelling of sagittal crest, I think the authors are actually talking about the nuchal crest here? The nuchal crest is the one that runs transversely across the skull and is widely visible in posterior view. The sagittal crest extends anteroposteriorly on the midline of the dorsal skull roof (the frontals and parietals) and visible in dorsal view, and maybe slightly in lateral and posterior views if it is tall enough.

In the Description, of pathology ii, I am having a hard time seeing this fragment of bone forced into the space between two alveoli. Could it not be just a random bit of bone that has broken off the specimen during burial/diagenesis and was lodged between the alveoli during diagenesis? Interpreting this as a postmortem break due to a bite may be a stretch.

In the Description of pathology iii, the authors state that “the lack of fluvial damage suggests this (feature) is postmortem.” I am not sure what is meant by this statement. First, fluvial damage would be postmortem. Second, how does the lack of fluvial damage help interpret this structure as a bite? I am also unsure about interpreting this as a bite more generally: this part of the dentary is very thin. Could this not just be a little bit of cracking/breakage caused during burial/diagenesis? I’m a little concerned that because there is other evidence of bite marks nearby that the authors may be over-interpreting this breakage.

I think the “Position of Elements in Field” section, which currently comes after the description of the various injuries, makes more sense closer to the beginning of the manuscript.

In the Discussion, I find it implausible that injuries to the heads of tyrannosaurids were caused by the feet of other tyrannosaurids during intraspecific combat/dominance matches. It’s not impossible, but this just doesn’t pass the sniff test. Were the feet of tyrannosaurids capable of kicking this way? I know this is outside of the scope of this paper, and the paragraph discussing this is in reference to a particular paper by Rothschild (2013), but I encourage the authors to think about how you would actually diagnose pathologies caused by a kick rather than by a bite. There must be differences. Some expansion here may be a good idea.

In the Discussion, the authors state that they “took impressions of the premaxillary and anterior maxillary teeth and anterior dentary teeth of a number of theropods (with foam) to examine the arrangement of their teeth to compare to the traces seen on the elements here.” However, these results are not presented in any detail. It is only stated that the marks on the tyrannosaurid skull do not match the “size and spacing between (the teeth) of small-bodied theropods”. I encourage the authors to present their results in much more detail, both statistically and visually. If the authors have compiled these data then they should present them rather than just allude to them in a single paragraph. It sounds like this provides good evidence that the bite maker was a big theropod, which is something important to clearly demonstrate.

·

Basic reporting

This manuscript would be of general interest to the readership, and of course of particular interest to dinosaur paleontologists and paleoethologists.

Experimental design

The marks of the specimen are sufficiently illustrated for this paper.

The foam "bite casts" described on p. 21 are not illustrated, but as their use was not critical to any analysis or conclusion in this paper, their non-inclusion serves to keep this manuscript brief.

Validity of the findings

The description, hypotheses, and conclusions of this study seem sound. The authors are cautious in outlining what part of their description are strongly supported, and what parts cannot be confidently identified (e.g., whether some marks are pre- or postmortem; whether the consumer might have be Daspletosaurus or some other tyrannosaurid; etc.).

Additional comments

I would strongly encourage its publications. In my read, there are no major issues to address, although a few minor corrections (mostly typos) and comments are given below:

Corrections & Comments
p. 1, line 17 “carnivorans” should be “carnivores”. Unless Hone & Tanke have found something really spectacular: the first evidence of placental Carnivora in the Cretaceous.

p. 4, line 66 Another case of tyrannosaurid face biting: Peterson et al. 2009, cited elsewhere in the manuscript.

p. 9, line 198 Curiously, the type of Daspletosaurus torosus shows apparent fusion between the left and right premaxillae (character 97 in Holtz (2001)) but subsequently regarded that that and other authors as an ontogenetic or pathological condition. Perhaps Daspletosaurus was particularly rough on its snout tip, or maybe the assessment that the fusion is pathological needs to be re-examined.

p. 10, line 217 “serious” should be “series”.

p. 19, line 435 The use of “animalian prey” isn’t clear here. Prey, by its very nature, has to be animal in nature; otherwise it is “fodder” or “browse” or what have you.

p. 18, line 442 Typo: “hypothesised” should be “hypothesized”.

·

Basic reporting

The paper is generally well-organized and presented. However, a few areas could be improved.

-The word “carnivorans” should be replaced with “carnivores” in the first sentence of the abstract to avoid confusion with the mammalian order Carnivora, where the term “carnivorans” is typically applied.

-The first sentence of the Introduction: omit commas after “ecology” and “interactions”.

-The last paragraph of the Introduction: this comment about the difference between the term “carnivore-consumed” and “predator-prey” is valid and should be included. However, it seems out of place here. Perhaps it could be re-worked in another way or mentioned in the discussion.

-Premortem: D. Replace “A serious” with “A series”.

-While it may be due to the lower-resolution of the draft images, many of the lesions discussed are not easily visible in the figures. For example, Figure 1 is very difficult to read due to the boldness, size, and placement of the arrows in relation to the size and distribution of lesions. The same is true for Figure 2.

-Figure 3 could be restructured to show closer images of the afflicted regions of interest; as currently imaged, the lesions are not as apparent as they could be (i.e., Figure 3 should be more like Figs. 4-7).

-Figure 17 should be revised for clarity and font sizes reduced.

Experimental design

The experimental design/study design is satisfactory.

Validity of the findings

The validity of the findings is satisfactory. However I did notice something with the discussion of Premortem: M. (lesions on the surangular).
The shape and orientation of the surangular lesions is consistent with Wolff et al., 2009, possibly the result of an infection rather than a bite (i.e. Trichomonas-like infection). However, since Trichonomas-like infections are spread today in avians through face-biting/pecking (among other vectors), a nice cause-and-effect inference could be discussed. *FYI – almost identical marks are present on the surangular of BMR P2002.4.1 (Jane) but have not been published (yet) in relation to the established facial lesions on that specimen (Peterson et al., 2009).

Additional comments

The paper does need a bit of polishing in terms of formatting (it could probably be a little more concise) but a very nice contribution!

---

## Round 0.3 · accepted · Accept

The manuscript is now acceptable. I have made a few editorial changes, and corrected remaining typos in an attached pdf.